# TMEM65 is a mitochondrial inner-membrane protein

Naotaka Nishimura[1], Tomomi Gotoh[1,2], Yuichi Oike[1] and Masato Yano[3]

[1] Department of Molecular Genetics, Graduate School of Medical Sciences, Kumamoto University, Kumamoto, Japan
[2] Department of School Health, Faculty of Education, Kumamoto University, Kumamoto, Japan
[3] Department of Medical Technology, Faculty of Health Sciences, Kumamoto Health Science University, Kumamoto, Japan

## ABSTRACT

It has been reported that the expression of TMEM65 is regulated by steroid receptor RNA activator (SRA). To date, however, the localization and function of TMEM65 remained unknown. We analyzed the intracellular localization of TMEM65. Immunoblot and immunostaining analysis revealed mitochondrial localization of TMEM65. Alkali extraction analysis and digitonin extraction test using isolated mitochondria revealed that TMEM65 is an integral membrane protein that localizes to the inner-membrane of mitochondria. Analysis using deletion mutants of TMEM65 suggested that the N-terminal region (1–20) of this protein is sufficient for mitochondrial targeting and that this mitochondrial targeting signal (MTS) is cleaved between the amino acid positions 35 and 64, which contain a putative recognition site of matrix processing protease (MPP). Together, these results suggest that TMEM65 is imported into the mitochondria, integrated into mitochondrial inner-membrane, and processed into its mature form by an MPP.

Corresponding author
Masato Yano,
yano@kumamoto-hsu.ac.jp

## INTRODUCTION

The mitochondrion is an organelle compartmented by two membranes (outer-membrane and inner-membrane) (*Pfanner & Wiedemann, 2002*), which divide this organelle into four compartments (outer-membrane, intermembrane space, inner-membrane and matrix). In the inner-membrane, the respiration complexes coupled with ATP synthase produce ATP by oxidative phosphorylation. The mitochondrial function is dynamically regulated by intracellular environmental changes such as energy demand (*Chan, 2006*; *Schapira, 2006*; *Nunnari & Suomalainen, 2012*). Most mitochondrial proteins are coded in the genomic DNA, synthesized in cytosol, and imported into mitochondria (*Koopman et al., 2013*). Some of these proteins are synthesized in the cytosol as precursors harboring mitochondrial targeting signals (MTS) at their N-terminus, which is then imported into the mitochondria and processed into mature form by a matrix processing protease (MPP) that cleaves and removes the MTS (*Pfanner & Wiedemann, 2002*; *Gakh, Cavadini & Isaya, 2002*).

Mitochondrial dysfunction leads to a variety of disease (*Schapira, 2006*), including Leigh syndrome. Deficiency of the respiration complex IV (cytochrome *c* oxidase) is commonly observed in Leigh syndrome patients (*Schapira, 2006*; *Huntsman et al., 2005*). Mitochondrial disorder enhances anaerobic energy production, thus increasing the concentration of lactic acid in the blood and leading to acidosis. Elevated blood lactic acid concentration is seen also in French–Canadian Leigh syndrome (LSFC) patients, who show delayed psychiatric and locomotive development and neurodegeneration in the basal ganglion (*Huntsman et al., 2005*). Studies have shown that LSFC is caused by mutations in the gene encoding leucine-rich pentatricopeptide repeat-containing protein (LRPPRC) (*Mootha et al., 2003*). LRPPRC regulates mitochondrial mRNA stability and the deficiency of this protein results in defective assembly of oxidative phosphorylation complexes due to the scarcity of mitochondrial mRNAs (*Sasarman et al., 2010*). It has also been reported that LRPPRC associates with the SRA stem-loop-interacting RNA-binding protein (SLIRP), forming a ribonucleoprotein complex (*Sasarman et al., 2010*). The steroid receptor RNA activator (SRA), a long non-coding RNA that regulates the activity of nuclear receptors, associates with SLIRP and regulates downstream target genes, including TMEM65 (*Foulds et al., 2010*). These findings suggested that TMEM65 likely regulates mitochondrial functions and contributes to the development of LSFC. However, the intracellular localization and functions of TMEM65 remained unknown.

In this study, we analyzed the subcellular localization of TMEM65. Analysis of the amino acid sequence of TMEM65 suggested that this protein might have a putative N-terminal MTS and three putative transmembrane regions. Results of immunoblot and immunostaining analyses indicated mitochondrial localization of TMEM65. Alkali extraction assay and digitonin extraction test using isolated mitochondria indicated that endogenous TMEM65 is localized to the mitochondrial inner-membrane. Analysis using deletion mutants of TMEM65 fused with enhanced green fluorescent protein (EGFP) suggested that the N-terminal (1–20) region of TMEM65 is sufficient for its mitochondrial targeting. We also found that the N-terminal MTS is cleaved off (processed) by MPP, and that the cleavage site is between amino acid residues at positions 35 and 64 of hTMEM65. Therefore, TMEM65 is imported into the mitochondria, integrated into the mitochondrial inner-membrane, and processed into mature form.

## MATERIALS AND METHODS

### Prediction analysis

The putative transmembrane regions of hTMEM65 were analyzed by using the PSPORT prediction program (http://psort.hgc.jp/form.html). The threshold was set at $-1.91$. The putative recognition site of mitochondrial MPP was also analyzed by PSPORT program.

### Reagents

All reagents were purchased from Sigma-Aldrich (Missouri, USA), Wako (Osaka, Japan), or Takara (Kyoto, Japan) unless otherwise stated. The plasmid pCMV6-Entry-hTMEM65 used for expression of human TMEM65 C-terminally-tagged with FLAG sequence

(hTMEM65-FLAG) was purchased from OriGene (Maryland, USA). An empty plasmid, pCMV6-Entry, was also obtained from OriGene.

## Construction of plasmid

The plasmids used for the expression of deletion mutants of hTMEM65 C-terminally-fused with EGFP were constructed using PCR method. The DNA fragments encoding the amino acids (1–20), (1–34), (1–64), (1–94) and (1–240) of hTMEM65 were amplified by PCR using primers containing *Eco* RI or *Bam* HI recognition site. The downstream primers used for constructing the plasmids encoding (1–20)hTMEM65-EGFP, (1–34)hTMEM65-EGFP, (1–64)hTMEM65-EGFP, (1–94)hTMEM65-EGFP, and p(1–240)hTMEM65-EGFP were 5′-AAAAAAGGATCCAACGGGCCCGGCCTCAG-3′, 5′-AAAAAAGGATCCAAGCAGCAGCACCAGGA-3′, 5′-AAAAAAGGATCCAACTCCATGGGCTCCTTC-3′, 5′-AAAAAAGGATCCAAGAAGCGGTGCAGCTC-3′, and 5′-AAAAAAGGATCCAGACTTTTCGTTTCCAGTTTTTC-3′, respectively. The upstream primer 5′-AAAAAAGAATTCATGTCCCGGCTGCTG-3′ was used in the construction of all plasmids. In each primer, the underlined sequence is *Eco* RI or *Bam* HI recognition site. The plasmid, pCMV6-Entry-hTMEM65, was used as a common template. Amplified DNA fragments were digested with *Eco* RI and *Bam* HI, purified, and then cloned into *Eco* RI/*Bam* HI site of pEGFP-N1 (Clontech, California, USA). Sequences of the cloned DNA fragments were confirmed by sequencing using an ABI PRISM 310 Genetic Analyzer (Applied Biosystems, California, USA) and BigDye Terminator v1.1 Cycle Sequencing Kit (Applied Biosystems).

## Cell culture and transfection

HeLa cells were cultured in growth medium [Dulbecco's Modified Eagle's Medium (DMEM) supplemented with 10% fetal calf serum] at 37 °C under a humidified atmosphere of 5% $CO_2$. Lipofectamine 2000 (Invitrogen, California, USA) was used for transfection of plasmids. After transfection, G418-resistant HeLa cells were cloned by limiting dilution, and were cultured for further analysis.

Small interfering RNAs (siRNAs) used for silencing the expression of hTMEM65 mRNA (siRNA1: 5′-GGAAUUAAGACAGUAACAGUAUAGA-3′, siRNA2: 5′-AGAUACAACAUCAGCGUAUGAGUGA-3′) were purchased from Origene, and were transfected into HeLa cells using Lipofectamine RNAi max (Invitrogen) and Opti-MEM (Gibco, New York, USA). The cells were cultured for 3 days and used for analysis.

## Immunoblot analysis

The HeLa cells were harvested with the help of phosphate-buffered saline (PBS) containing 1 mM EDTA and were washed twice with PBS. The cells were lysed using PBS containing 1% Triton X-100 or (X2) SDS-PAGE loading buffer [50 mM Tris/HCl (pH 7.5), 2% SDS, 40% glycerol, 0.02% bromophenol blue (BPB), 2% 2-mercaptoethanol]. After centrifuging at $10,000 \times g$ for 5 min to remove debris, the proteins present in the lysate were separated by SDS-PAGE and transferred onto nitrocellulose membrane. After blocking with 5% skim milk, the membrane was used for immunoblot analysis using ECL Western Blotting Detection Reagents (GE Healthcare, Buckinghamshire, UK) as described previously

(*Yano et al., 2000*). Immunoblotting was performed by using anti-TMEM65 antibody (Sigma-Aldrich), anti-Hsc70 antibody (Santa Cruz Biotechnology, California, USA), anti-porin antibody (Calbiochem, California, USA), anti-Hsp60 antibody (Enzo Life Science, New York, USA), anti-acetyl histone H3 (K9/K14) antibody (Cell signaling, Massachusetts, USA), anti-calnexin antibody (Enzo Life Science), anti-ABCB10 antibody (Proteintech, Illinois, USA), anti-FLAG antibody (Wako), and anti-GFP antibody (Medical & Biological Laboratories, Nagoya, Japan).

## Double staining for hTMEM65 and mitochondria-targeted DsRed

The plasmid encoding hTMEM65-FLAG and that encoding *Discosoma sp*. red fluorescent protein (DsRed) fused with MTS sequence of pre-ornithine transcarbamylase (MtDsRed) (*Gotoh et al., 2004*), were co-transfected into COS-7 cells using Lipofectamine LTX (Invitrogen) and Opti-MEM (Gibco). After transfection, the cells were cultured for 24 h, fixed using 4% paraformaldehyde and then with methanol. Following this, the cells were washed with PBS, incubated with anti-FLAG antibody, and treated with Alexa Fluor 488-conjugated secondary antibody. Fluorescence images of labeled cells owing to DsRed and Alexa Fluor 488 dye were acquired using a Fluoview FV300 fluorescent microscope (Olympus, Tokyo, Japan).

Alternatively, the plasmids encoding hTMEM65 deletion mutants C-terminally-fused with EGFP and the plasmid encoding MtDsRed, were co-transfected into COS-7 cells, and the fluorescence of EGFP and DsRed were observed under the fluorescence microscope.

## Subcellular fractionation

HeLa cells were harvested and washed twice with PBS. The cells were suspended in the isolation buffer A [3 mM Hepes-KOH (pH 7.4), 0.21 M mannitol, 0.07 M sucrose, 0.2 mM EGTA] and homogenized with a Dounce homogenizer (Wheaton, Illinois, USA) on ice. The homogenate was overlaid on 0.34 M sucrose and was then centrifuged at $500 \times g$. The supernatant containing mitochondria and microsome was collected. This step was repeated two more times. The supernatant was further centrifuged at $10,000 \times g$, to yield mitochondrial fraction as pellet. After repeating the centrifugation step four times, the supernatant was centrifuged at $100,000 \times g$, to yield microsomal fraction as pellet and cytosolic fraction as supernatant. The nuclear fraction was isolated by following the methods described previously (*Schreiber et al., 1989*).

## Fractionation of mitochondria

The isolated mitochondria were resuspended in alkali extraction reagent [0.1 M $Na_2CO_3$, 0.02 M $NaHCO_3$ (pH 10.5)] and sonicated. The suspension was then centrifuged at $100,000 \times g$, and the supernatant was collected as mitochondrial soluble fraction. The pellet, an alkali-insoluble fraction, was collected as the mitochondrial membrane fraction.

To separate mitochondrial outer-membrane and mitoplast (inner-membrane plus matrix), isolated mitochondria were resuspended in 0.15 mg/mL digitonin and centrifuged at $10,000 \times g$. The resulting supernatant was collected as outer-membrane fraction, and the pellet was recovered as mitoplast (*Pallotti & Lenaz, 2007*).

## RESULTS

### Intracellular localization of hTMEM65

It has been reported that reduced expression of SRA is associated with attenuated TMEM65 mRNA expression (*Foulds et al., 2010*). However, both the function and intracellular localization of TMEM65 remained unknown. Therefore, we examined the intracellular localization of TMEM65.

Results of the PSPORT prediction analysis suggested that human TMEM65 (hTMEM65, 240 amino acids) had three putative transmembrane regions (Fig. 1A) and a putative recognition site (RRL | GT between the amino acid residues 52–56 of hTMEM65) of mitochondrial MPP (R-2 motif, XRX | X(S/X)) (*Gakh, Cavadini & Isaya, 2002*). Immunoblot analysis of the cell lysate prepared from HeLa cells overexpressing FLAG-tagged hTMEM65 (hTMEM65-FLAG, 271 amino acids) using anti-FLAG and anti-TMEM65 antibodies showed an intense, approximately 24-kDa band, likely corresponding to hTMEM65-FLAG (Fig. 1B). Because the observed approximately 24-kDa protein contained a C-terminal FLAG tag of approximately 3 kDa, the molecular size of the hTMEM65 was estimated to be about 21 kDa. The molecular weight of hTMEM65, measured according to its deduced amino acid sequence, was approximately 26 kDa. To verify the size of endogenous hTMEM65, we silenced its expression using siRNAs. Immunoblot analysis using anti-TMEM65 antibody showed that the band observed in the control sample (approximately 21 kDa) was absent in the samples prepared from the cells treated with siRNAs, suggesting that the approximately 21-kDa band corresponded to endogenous TMEM65 (Fig. 1C). Therefore, we hypothesized that TMEM65 might be post-translationally processed into a mature form. To examine the intracellular localization of TMEM65, HeLa cells were fractionated and the fractions were subjected to immunoblot analysis (Fig. 1D). Anti-histone, anti-Hsc70, anti-porin, and anti-calnexin antibodies were used to detect nuclear, cytosolic, mitochondrial, and microsomal fractions, respectively. When anti-TMEM65 antibody was used, an approximately 21-kDa major band was detected in mitochondrial fraction, which indicated that TMEM65 was a mitochondrial protein. These results suggested that hTMEM65 was likely synthesized as an approximately 26-kDa precursor, which was then processed into a mature form (approximately 21 kDa) in mitochondria. Indeed, hTMEM65 contains a putative N-terminal MTS (approximately 5 kDa) that was probably cleaved and removed during the mitochondrial import (see Fig. 1A).

### Intra-mitochondrial localization of hTMEM65

The localization of TMEM65 was further examined by double staining with the help of hTMEM65-FLAG and mitochondria-targeted DsRed (MtDsRed) (Fig. 2A). In COS-7 cells expressing both hTMEM65-FLAG and MtDsRed, green fluorescence (Alexa Fluor 488) from hTMEM65-FLAG and red fluorescence (MtDsRed) from the mitochondria were merged. This result confirmed that TMEM65 was a mitochondrial protein. To examine whether TMEM65 was a soluble protein or a membrane protein, we performed alkali-extraction analysis using isolated mitochondria and subjected the fractions to immunoblot

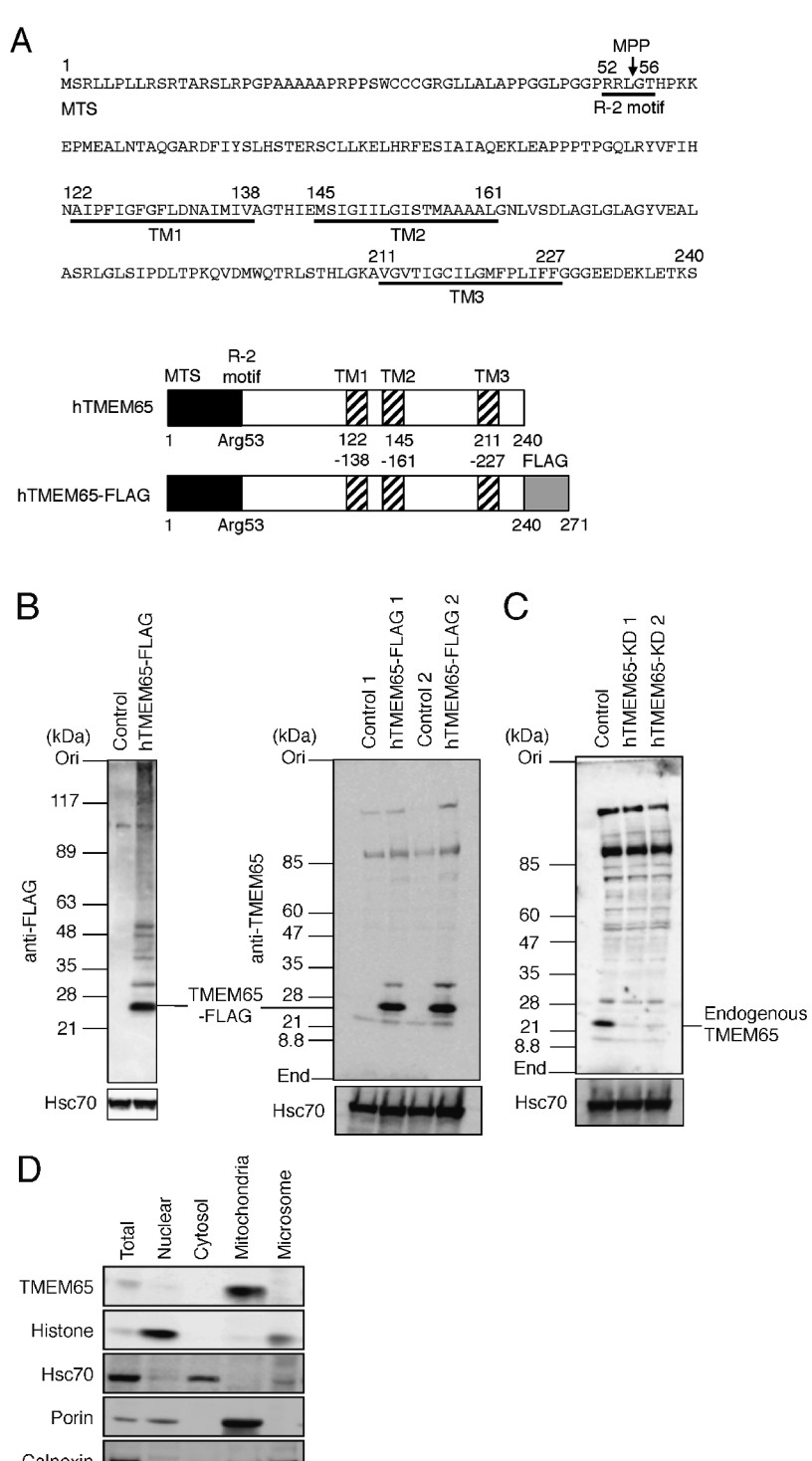

**Figure 1 Analysis of intracellular localization of hTMEM65.** (A) The amino acid sequence of hT-MEM65 (top panel), and the simple structures of hTMEM65 (approximately 26 kDa) and that of the C-terminally-fused with FLAG tag (hTMEM65-FLAG (approximately 29 kDa)) (bottom panel) are shown. The putative recognition site (R-2 motif) of MPP and putative transmembrane (TM) regions of hTMEM65 predicted by PSPORT are shown. 

**Figure 1 (...continued)**

Arrow indicates putative MPP cleavage site. The amino acid positions corresponding to each putative TM region are as follows: TM1, 122–138; TM2, 145–161; TM3, 211–227. The filled boxes indicate the putative MTS (approximately 5 kDa). The hatched boxes indicate putative TM regions. The shaded box indicates the C-terminal region containing FLAG tag (approximately 3 kDa). (B) Total proteins were extracted from HeLa cells stably transfected with empty plasmid (control) or hTMEM65-FLAG-expressing plasmid (hTMEM65-FLAG), and subjected to SDS-PAGE followed by immunoblot analysis using anti-FLAG antibody (left panel) and anti-TMEM65 antibody (right panel). (C) Scrambled siRNA (control) or hTMEM65-siRNAs (hTMEM65-KD) were transfected into HeLa cells. After 3 days of culture, the cells were harvested, lysed, and the lysates were subjected to immunoblot analysis using anti-TMEM65 antibody. (D) HeLa cells were fractionated into each organelle, and the fractions were analyzed by immunoblotting.

analysis (Fig. 2B). TMEM65, as well as porin (a mitochondrial outer-membrane protein), was present in the alkali-resistant fraction that contained mitochondrial membrane proteins, whereas Hsp60 (a soluble protein in the mitochondrial matrix) was present in alkali-extracted fraction. This result indicated that TMEM65 was a mitochondrial protein integrated in the membrane. We further analyzed whether TMEM65 was localized to the mitochondrial inner-membrane or outer-membrane by digitonin extraction method (Fig. 2C). TMEM65, as well as ABCB10 (a mitochondrial inner-membrane protein), was presented in the fraction that contained mitochondrial inner-membrane and matrix, whereas porin was found in the mitochondrial outer-membrane fraction (*Chen et al.,* *2009*). These results clearly suggested that TMEM65 was a mitochondrial inner-membrane protein.

## Analysis of MTS of hTMEM65

Most of the mitochondrial proteins are synthesized in the cytosol with an MTS and then imported into the mitochondria. A number of mitochondrial proteins are synthesized with N-terminal MTS rich in positively charged residues such as arginine (Arg). The MTS is cleaved and removed (processed) by MPP, which yields a mature protein, and is accompanied by mitochondrial import. Because the putative amino acid sequence of TMEM65 contained many positively charged residues in its N-terminus, we hypothesized that TMEM65 might possess an N-terminal MTS, and that the signal sequence might be cleaved and removed by proteases to produce a mature TMEM65 (see Fig. 1A). To investigate this possibility, we constructed deletion mutants of TMEM65 C-terminally-fused with EGFP (Fig. 3A). These mutants were co-expressed with MtDsRed in COS-7 cells, and the intracellular localization was observed by using a fluorescent microscope (Fig. 3B). The mutants with N-terminal (1–20), (1–34), (1–64), (1–94) and (1–240) regions of the hTMEM65 C-terminally-fused with EGFP were localized to the mitochondria together with MtDsRed, whereas EGFP was localized to the cytosol, indicating that the (1–20) region of hTMEM65 was sufficient for mitochondrial targeting. We next examined whether the N-terminal MTS (approximately 5 kDa) of hTMEM65 was processed during mitochondrial import of this protein (Fig. 3C, also see Fig. 1A). The mutants were expressed in COS-7 cells and the expressed proteins were analyzed by immunoblotting. A band, likely corresponding to the unprocessed protein, was observed in the lysates

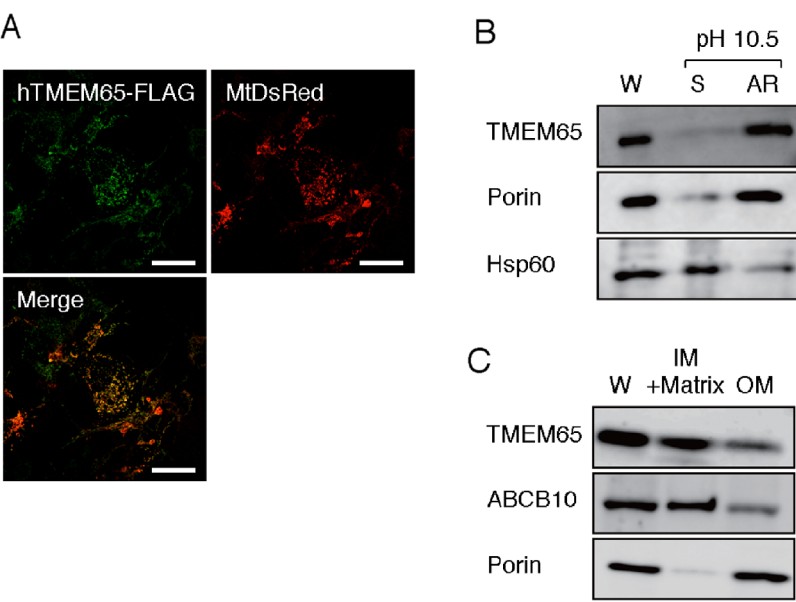

**Figure 2 Analysis of intra-mitochondrial localization of hTMEM65.** (A) COS-7 cells were co-transfected with a plasmid encoding hTMEM65-FLAG and a plasmid encoding MtDsRed. After 24 h of culture, the cells were subjected to immunostaining using anti-FLAG antibody and an Alexa Fluor 488-conjugated secondary antibody. Fluorescence due to Alexa Fluor 488 (hTMEM65-FLAG) or MtDsRed was photographed. The merged image is also shown (Merge). Scale bar, 50 μm. (B) Mitochondria were isolated from HeLa cells and were subjected to alkali extraction. The whole mitochondria (W), soluble fractions (S), and alkali-resistant fractions (AR) were subjected to immunoblot analysis using anti-TMEM65 antibody. Mitochondrial porin (an outer-membrane protein) and Hsp60 (a matrix protein) were also immunostained as controls. (C) Mitochondria isolated from HeLa cells were subjected to digitonin extraction to separate outer-membrane fraction (OM) and the fraction containing inner-membrane and matrix (IM+Matrix). These fractions and whole mitochondria (W), were subjected to immunoblot analysis using anti-TMEM65 antibody. Mitochondrial porin (an outer membrane protein) and ABCB10 (an inner-membrane protein) were also immunostained as controls.

prepared from cells expressing (1–20)hTMEM65-EGFP or (1–34)hTMEM65-EGFP (indicated by asterisks in Fig. 3C). A second band corresponding to that of EGFP, likely due to the degradation of the mutants owing to the instability of the unprocessed MTS region, was also observed. Notably, lysates prepared from cells expressing (1–64)hTMEM65-EGFP or (1–94)hTMEM65-EGFP showed a band likely corresponding to the processed protein (indicated with hash marks in Fig. 3C). However, in these blots, a band corresponding to EGFP was not detected, suggesting that the processed proteins were stable in the mitochondria. These results suggested that MPP recognition site was present in hTMEM65 between the amino acid residues at position 35 and 64.

## DISCUSSION

Previous reports suggested the possibility that TMEM65 might participate in the regulation of mitochondrial function (*Foulds et al., 2010*). However, the localization and function of TMEM65 were unknown. Therefore, we examined the localization of TMEM65 and found that TMEM65 is a mitochondrial inner-membrane protein containing an N-terminal MTS. Because the calculated molecular weight of TMEM65 was approximately

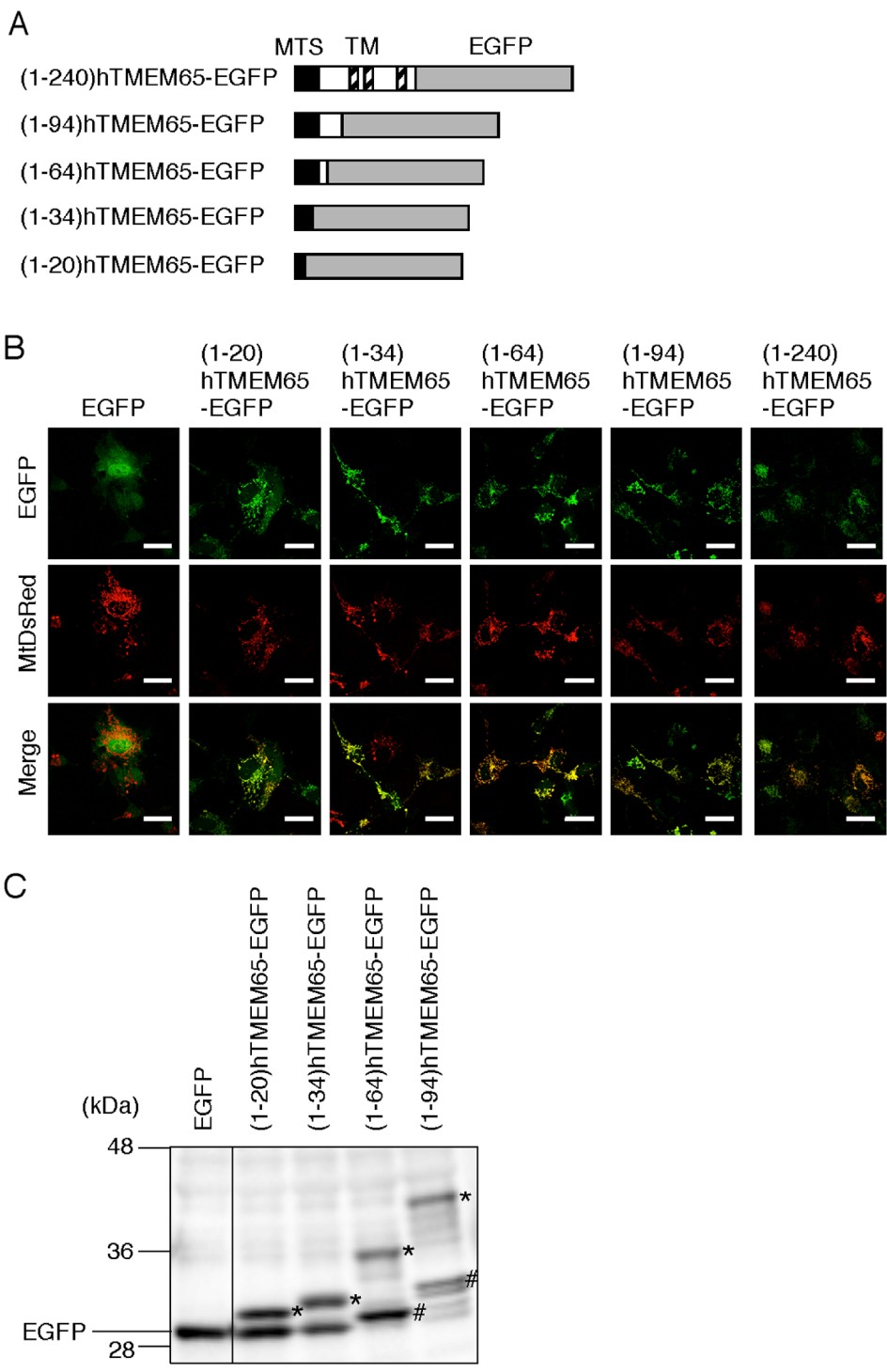

**Figure 3  Analysis of the MTS of hTMEM65.** (A) The structures of the deletion mutants of hTMEM65 C-terminally-fused with EGFP (hTMEM65-EGFPs) are shown. Filled box indicates putative MTS. Hatched box indicates the putative transmembrane (TM) regions of hTMEM65. Shaded box indicates EGFP. (B) Deletion mutants of hTMEM65 fused with EGFP shown in A and MtDsRed were co-expressed in COS-7 cells, and the fluorescence from hTMEM65-EGFPs or MtDsRed was photographed. Merged images are also shown (Merge). Scale bar, 50 μm.  

**Figure 3 (...continued)**

(C) Deletion mutants of hTMEM65 fused with EGFP shown in A were expressed in COS-7 cells, and the proteins extracted from the cells were subjected to immunoblot analysis using anti-GFP antibody. The bands likely corresponding to unprocessed hTMEM65-EGFPs are indicated with asterisks. The bands likely corresponding to the processed hTMEM65-EGFPs are indicated with hash marks.

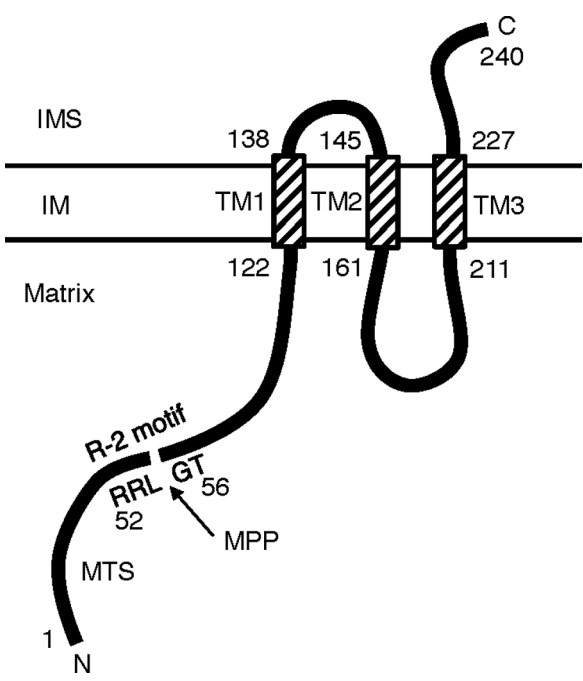

**Figure 4 Putative topology of hTMEM65.** The putative MPP recognition site (R-2 motif in the amino acid sequence 52–56 (RRL | GT)) and three putative transmembrane regions (TM, hatched boxes) of TMEM65 predicted by PSPORT program, are shown. The amino acid positions corresponding to the putative TM regions are as follows: TM1, 122–138; TM2, 145–161; TM3, 211–227. IMS, intermembrane space; IM, inner-membrane.

26 kDa and the observed size of endogenous TMEM65 was approximately 21 kDa, we hypothesized that TMEM65 contained an MTS of approximately 5 kDa, which would be processed by a MPP. Indeed, the region around Arg53 of TMEM65 had an R-2 motif (XRX | X(S/X)), a putative recognition site of MPP (*Gakh, Cavadini & Isaya, 2002*), as revealed by the prediction analysis using PSPORT. Analysis using deletion mutants of TMEM65 fused with EGFP demonstrated that the 20 N-terminal amino acids (1–20) were sufficient for mitochondrial targeting, and that the recognition site of the MPP was between the amino acid residues at positions 35 and 64 of hTMEM65. Thus, TMEM65 harbored an N-terminal MTS that was likely cleaved at the R-2 motif around Arg53 by an MPP. Since TMEM65 contains three putative transmembrane regions, it is highly likely that this protein is an inner-membrane protein, in which the N-terminal end would be exposed to matrix and the C-terminal end might be exposed to intermembrane space (Fig. 4).

In this study, we were unable to identify the function of TMEM65. However, our present results will contribute to future functional analysis of TMEM65. Further analysis may reveal the functional relationships between TMEM65, SRA, SLIRP, and LRPPRC that contribute to the pathological processes in LSFC.

**Abbreviations**

| | |
|---|---|
| **ABCB10** | ATP-binding cassette transporter 10 |
| **DsRed** | *Discosoma sp*. red fluorescent protein |
| **EGFP** | Enhanced green fluorescent protein |
| **Hsc70** | Heat shock cognate protein 70 |
| **Hsp60** | Heat shock protein 60 |
| **LRPPRC** | Leucine-rich pentatricopeptide repeat containing protein |
| **LSFC** | French–Canadian type of Leigh syndrome |
| **MPP** | Matrix processing protease |
| **MTS** | Mitochondrial targeting signal |
| **PBS** | Phosphate-buffered saline |
| **siRNA** | Small interfering RNA |
| **SLIRP** | SRA stem-loop-interacting RNA-binding protein |
| **SRA** | Steroid receptor RNA activator |
| **TMEM65** | Transmembrane protein 65 |

## ACKNOWLEDGEMENTS

We thank our colleagues for valuable suggestions and discussion.

### Funding

This work was supported by the grants-in-aid 23590340 (to MY), 23590365 (to TG), and 22116009 (to YO) from the Ministry of Education, Science, Technology, Sports and Culture of Japan. The funders had no role in study design, data collection and analysis, decision to publish, or preparation of the manuscript.

### Grant Disclosures

The following grant information was disclosed by the authors:
The Ministry of Education, Science, Technology, Sports and Culture of Japan: 23590340, 23590365, 22116009.

### Competing Interests

The authors declare there are no competing interests.

## Author Contributions

- Naotaka Nishimura and Masato Yano conceived and designed the experiments, performed the experiments, analyzed the data, contributed reagents/materials/analysis tools, wrote the paper, prepared figures and/or tables, reviewed drafts of the paper.
- Tomomi Gotoh analyzed the data, contributed reagents/materials/analysis tools, reviewed drafts of the paper.
- Yuichi Oike contributed reagents/materials/analysis tools.

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
