# Peer review of "TMEM65 is a mitochondrial inner-membrane protein"

_PeerJ, doi:10.7717/peerj.349_

## Round 0.1 · original submission · Major Revisions

Editor's comments:
Generally, the manuscript is of valuable for documentation in the specific field. However, the criticisms raised by the external Rev-2 and 3 should carefully be addressed with the additional experiment. In parallel, please check the English Usage before resubmission.
Thanks
Regards

Reviewer 1 ·

Basic reporting

The paper is on the whole well written and straightforward to understand, some light editing for spelling and grammar will be easily carried out. There is enough information and literature cited in the introduction and the methods section to understand the background of the manuscript, and figures are appropriate.

Experimental design

The methods used are well established in the field and directly address the questions being asked. There is more than enough detail in the methods, figure legends and results sections, that the experiments can be reproduced.Relevant controls are included for all of the experiments.

Validity of the findings

The manuscript Nishimura et al. describes the characterization of TMEM65 as a mitochondrial protein. A variety of techniques are used to show that it is located in the mitochondrial inner membrane as a transmembrane protein, and that its N-terminal region is responsible for its mitochondrial targeting. The experiments are carefully and correctly performed, with all of the appropriate controls in place. It is noted that the deletion mutant experiments in Fig. 3A-C show that the N-terminal region is sufficient for mitochondrial targeting, but the experiment showing the converse has not been done, that is deletion of the N-terminal region to demonstrate that it is necessary for mitochondrial targeting. However, the apparent transmembrane domains of TMEM65 could contain targeting information on their own, complicating the result, and the authors’ conclusions can be made without such an experiment. I would recommend that the putative topology of TMEM65 in the inner membrane be presented in the discussion, and possibly in a diagram figure, as the results strongly suggest the N-terminus is in the matrix (cleaved by MPP) and there seems agreement on three transmembrane domains. The manuscript can therefore be recommended for publication.

Reviewer 2 ·

Basic reporting

The article is well structured. However, the English requires definitely some polishing and editing; e.g. the first sentence in the abstract (TMEM65 is a supposed protein that co-relates with steroid receptor RNA activator (SRA)) is incomprehensible. Often the text lacks articles.

Experimental design

It addresses and answers a distinct question, the intracellular localization of the protein TMEM65. Overall, the methods are adequate (see also below).

Validity of the findings

The authors experimentally show and correctly conclude that TMEM65 is a mitochondrial inner membrane protein; a conclusion expected from the protein sequuence. The authors state that analysis of aa sequence suggest a N-terminal targeting signal and three transmembrane segments. These results including the programmes used for the analysis (Mitoprot? Claros et al?, TMHMM Server v. 2.0 ?) should be stated in the text. It would be helpful to show the aa sequence with marked features (targeting signal, TM segment, processing site) in Fig. 1A. The data concerning the localisation are robust. It is also very likely that processing occurs, although it is not formally shown. The experiment presented in Fig. 3D is not technically solid and the data are not convincing. The gel shows a smear of overexpressed protein, but no distinct bands representing specific complexes. It cannot be concluded that TMEM65 forms a larger complex. Here, the quality of the gel should be improved, endogenous protein should be analysed and a control for the running behaviour of the monomeric TMEM65 should be added (e.g. recombinant TMEM65). Otherwise, the data and the conclusion has to be removed. Without further investigation about the nature of the complex, this part does not add much information, anyway.

Reviewer 3 ·

Basic reporting

No comments

Experimental design

No comments

Validity of the findings

No comments

Additional comments

This is an initial characterization of TMEM65, which could be a target gene for the regulation by the causative gene product for Leigh syndrome, with the emphasis on its cellular localization and membrane association. The authors showed that TMEM65 is localized in mitochondria, is an integral membrane protein, and is probably associated with the inner mitochondrial membrane. They also showed that the N-terminal 20 residues of TMEM65 could carry EGFP to mitochondria. Most of the experiments were well controlled and the obtained results were properly presented except for the ones pointed out below.

Now the problem to the reviewer is that the overall results contained in this manuscript appear too thin. In other words, this manuscript is too premature as it stands now. Perhaps, the authors can perform in vitro import of RI-labeled TMEM65 (and its derivatives) into isolated mitochondria. In vitro import can help the authors to claim more on the properties of TMEM65 and strengthen their claim in this manuscript. For example, although Fig. 3C is not convincing enough to show that TMEM65 has a cleavable presequence, in vitro import will easily reveal the presence of such a presequence. It is also interesting to see if in vitro imported TMEM65 can be assembled into the large complex on BN-PAGE gel. The quality of Fig. 3D is too poor to demonstrate that TMEM65 forms a large complex in the mitochondrial inner membrane, but BN-PAGE of RI-labeled TMEM65 will offer clearer results.

---

## Round 0.2 · accepted · Accept

Your revised version has carefully been rechecked for its suitability with regard to the general documentation in this journal, The Peer J. I have found that your resubmission is should be published, and the functional analysis of the mitochondrial membrane-localized TMEM65 is further subject to uncover. It means this initial stage of the publication will give some new information of the protein.